# MULTI-TIME ATTENTION NETWORKS FOR IRREGULARLY SAMPLED TIME SERIES

**Satya Narayan Shukla & Benjamin M. Marlin**
College of Information and Computer Sciences
University of Massachusetts Amherst
Amherst, MA 01003, USA
{snshukla,marlin}@cs.umass.edu

## ABSTRACT

Irregular sampling occurs in many time series modeling applications where it presents a significant challenge to standard deep learning models. This work is motivated by the analysis of physiological time series data in electronic health records, which are sparse, irregularly sampled, and multivariate. In this paper, we propose a new deep learning framework for this setting that we call *Multi-Time Attention Networks*. Multi-Time Attention Networks learn an embedding of continuous time values and use an attention mechanism to produce a fixed-length representation of a time series containing a variable number of observations. We investigate the performance of this framework on interpolation and classification tasks using multiple datasets. Our results show that the proposed approach performs as well or better than a range of baseline and recently proposed models while offering significantly faster training times than current state-of-the-art methods.[1]

## 1 INTRODUCTION

Irregularly sampled time series occur in application domains including healthcare, climate science, ecology, astronomy, biology and others. It is well understood that irregular sampling poses a significant challenge to machine learning models, which typically assume fully-observed, fixed-size feature representations (Marlin et al., 2012; Yadav et al., 2018). While recurrent neural networks (RNNs) have been widely used to model such data because of their ability to handle variable length sequences, basic RNNs assume regular spacing between observation times as well as alignment of the time points where observations occur for different variables (i.e., fully-observed vectors). In practice, both of these assumptions can fail to hold for real-world sparse and irregularly observed time series. To respond to these challenges, there has been significant progress over the last decade on building and adapting machine learning models that can better capture the structure of irregularly sampled multivariate time series (Li & Marlin, 2015; 2016; Lipton et al., 2016; Futoma et al., 2017; Che et al., 2018; Shukla & Marlin, 2019; Rubanova et al., 2019).

In this work, we introduce a new model for multivariate, sparse and irregularly sampled time series that we refer to as *Multi-Time Attention networks* or mTANs. mTANs are fundamentally continuous-time, interpolation-based models. Their primary innovations are the inclusion of a learned continuous-time embedding mechanism coupled with a time attention mechanism that replaces the use of a fixed similarity kernel when forming representation from continuous time inputs. This gives mTANs more representational flexibility than previous interpolation-based models (Shukla & Marlin, 2019).

Our approach re-represents an irregularly sampled time series at a fixed set of reference points. The proposed time attention mechanism uses reference time points as queries and the observed time points as keys. We propose an encoder-decoder framework for end-to-end learning using an mTAN module to interface with given multivariate, sparse and irregularly sampled time series inputs. The encoder takes the irregularly sampled time series as input and produces a fixed-length latent representation over a set of reference points, while the decoder uses the latent representations to produce reconstructions conditioned on the set of observed time points. Learning uses established methods for variational autoencoders (Rezende et al., 2014; Kingma & Welling, 2014).

---

[1]Implementation available at : https://github.com/reml-lab/mTAN

The main contributions of the mTAN model framework are: (1) It provides a flexible approach to modeling multivariate, sparse and irregularly sampled time series data (including irregularly sampled time series of partially observed vectors) by leveraging a time attention mechanism to learn temporal similarity from data instead of using fixed kernels. (2) It uses a temporally distributed latent representation to better capture local structure in time series data. (3) It provides interpolation and classification performance that is as good as current state-of-the-art methods or better, while providing significantly reduced training times.

## 2 RELATED WORK

An irregularly sampled time series is a time series with irregular time intervals between observations. In the multivariate setting, there can also be a lack of alignment across different variables within the same multivariate time series. Finally, when gaps between observation times are large, the time series is also considered to be sparse. Such data occur in electronic health records (Marlin et al., 2012; Yadav et al., 2018), climate science (Schulz & Stattegger, 1997), ecology (Clark & Bjørnstad, 2004), biology (Ruf, 1999), and astronomy (Scargle, 1982). It is well understood that such data cause significant issues for standard supervised machine learning models that typically assume fully observed, fixed-size feature representations (Marlin et al., 2012).

A basic approach to dealing with irregular sampling is fixed temporal discretization. For example, Marlin et al. (2012) and Lipton et al. (2016) discretize continuous-time observations into hour-long bins. This has the advantage of simplicity, but requires ad-hoc handling of bins with more than one observation and results in missing data when bins are empty.

The alternative to temporal discretization is to construct models with the ability to directly use an irregularly sampled time series as input. Che et al. (2018) present several methods based on gated recurrent unit networks (GRUs, Chung et al. (2014)), including an approach that takes as input a sequence consisting of observed values, missing data indicators, and time intervals since the last observation. Pham et al. (2017) proposed to capture time irregularity by modifying the forget gate of an LSTM (Hochreiter & Schmidhuber, 1997), while Neil et al. (2016) introduced a new time gate that regulates access to the hidden and cell state of the LSTM. While these approaches allow the network to handle event-based sequences with irregularly spaced vector-valued observations, they do not support learning directly from vectors that are partially observed, which commonly occurs in the multivariate setting because of lack of alignment of observation times across different variables.

Another line of work has looked at using observations from the future as well as from the past for interpolation. Yoon et al. (2019) and Yoon et al. (2018) presented an approach based on the multi-directional RNN (M-RNN) that can leverage observations from the relative past and future of a given time point. Shukla & Marlin (2019) proposed the interpolation-prediction network framework, consisting of several semi-parametric RBF interpolation layers that interpolate multivariate, sparse, and irregularly sampled input time series against a set of reference time points while taking into account all observed data in a time series. Horn et al. (2020) proposed a set function-based approach for classifying time-series with irregularly sampled and unaligned observation.

Chen et al. (2018) proposed a variational auto-encoder model (Kingma & Welling, 2014; Rezende et al., 2014) for continuous time data based on the use of a neural network decoder combined with a latent ordinary differential equation (ODE) model. They model time series data via a latent continuous-time function that is defined via a neural network representation of its gradient field. Building on this, Rubanova et al. (2019) proposed a latent ODE model that uses an ODE-RNN model as the encoder. ODE-RNNs use neural ODEs to model the hidden state dynamics and an RNN to update the hidden state in the presence of a new observation. De Brouwer et al. (2019) proposed GRU-ODE-Bayes, a continuous-time version of the Gated Recurrent Unit (Chung et al., 2014). Instead of the encoder-decoder architecture where the ODE is decoupled from the input processing, GRU-ODE-Bayes provides a tighter integration by interleaving the ODE and the input processing steps.

Several recent approaches have also used attention mechanisms to model irregularly sampled time series (Song et al., 2018; Tan et al., 2020; Zhang et al., 2019) as well as medical concepts (Peng et al., 2019; Cai et al., 2018). Most of these approaches are similar to Vaswani et al. (2017) where they replace the positional encoding with an encoding of time and model sequences using self-attention.

However, instead of adding the time encoding to the input representation as in Vaswani et al. (2017), they concatenate it with the input representation. These methods use a fixed time encoding similar to the positional encoding of Vaswani et al. (2017). Xu et al. (2019) learn a functional time representation and concatenate it with the input event embedding to model time-event interactions.

Like Xu et al. (2019) and Kazemi et al. (2019), our proposed method learns a time representation. However, instead of concatenating it with the input embedding, our model learns to attend to observations at different time points by computing a similarity weighting using only the time embedding. Our proposed model uses the time embedding as both the queries and keys in the attention formulation. It learns an interpolation over the query time points by attending to the observed values at key time points. Our proposed method is thus similar to kernel-based interpolation, but learning the time attention based similarity kernel gives our model more flexibility compared to methods like that of Shukla & Marlin (2019) that use similarity kernels with fixed functional forms. Another important difference relative to many of these previous methods is that our proposed approach attends only to the observed data dimensions at each time point and hence does not require a separate imputation step to handle vector valued observations with an arbitrary collection of dimensions missing at any given time point.

## 3   THE MULTI-TIME ATTENTION MODULE

In this section, we present the proposed Multi-Time Attention Module (mTAN). The role of this module is to re-represent a sparse and irregularly sampled time series in a fixed-dimensional space. This module uses multiple continuous-time embeddings and attention-based interpolation. We begin by presenting notation followed by the time embedding and attention components.

**Notation:** In the case of a supervised learning task, we let $\mathcal{D} = \{(\mathbf{s}_n, y_n) | n = 1, ..., N\}$ represent a data set containing $N$ data cases. An individual data case consists of a single target value $y_n$ (discrete for classification), as well as a $D$-dimensional, sparse and irregularly sampled multivariate time series $\mathbf{s}_n$. Different dimensions $d$ of the multivariate time series can have observations at different times, as well as different total numbers of observations $L_{dn}$. Thus, we represent time series $d$ for data case $n$ as a tuple $\mathbf{s}_{dn} = (\mathbf{t}_{dn}, \mathbf{x}_{dn})$ where $\mathbf{t}_{dn} = [t_{1dn}, ..., t_{L_{dn}dn}]$ is the list of time points at which observations are defined and $\mathbf{x}_{dn} = [x_{1dn}, ..., x_{L_{dn}dn}]$ is the corresponding list of observed values. In the case of an unsupervised task such as interpolation, each data case consists of a multivariate time series $\mathbf{s}_n$ only. We drop the data case index $n$ for brevity when the context is clear.

**Time Embedding:** Time attention module is based on embedding continuous time points into a vector space. We generalize the notion of a positional encoding used in transformer-based models to continuous time. Time attention networks simultaneously leverage $H$ embedding functions $\phi_h(t)$, each outputting a representation of size $d_r$. Dimension $i$ of embedding $h$ is defined as follows:

$$\phi_h(t)[i] = \begin{cases} \omega_{0h} \cdot t + \alpha_{0h}, & \text{if} \quad i = 0 \\ \sin(\omega_{ih} \cdot t + \alpha_{ih}), & \text{if} \quad 0 < i < d_r \end{cases} \tag{1}$$

where the $\omega_{ih}$'s and $\alpha_{ih}$'s are learnable parameters. The periodic terms can capture periodicity in time series data. In this case, $\omega_{ih}$ and $\alpha_{ih}$ represent the frequency and phase of the sine function. The linear term, on the other hand, can capture non-periodic patterns dependent on the progression of time. For a given difference $\Delta$, $\phi_h(t + \Delta)$ can be represented as a linear function of $\phi_h(t)$.

Learning the periodic time embedding functions is equivalent to using a one-layer fully connected network with a sine function non-linearity to map the time values into a higher dimensional space. By contrast, the positional encoding used in transformer models is defined only for discrete positions. We note that our time embedding functions subsume positional encodings when evaluated at discrete positions.

**Multi-Time Attention:** The time embedding component described above takes a continuous time point and embeds it into $H$ different $d_r$-dimensional spaces. In this section, we describe how we leverage time embeddings to produce a continuous-time embedding module for sparse and irregularly sampled time series. This *multi-time attention* embedding module mTAN$(t, \mathbf{s})$ takes as input a query time point $t$ and a set of keys and values in the form of a $D$-dimensional multivariate sparse and irregularly sampled time series $\mathbf{s}$ (as defined in the notation section above), and returns a $J$-

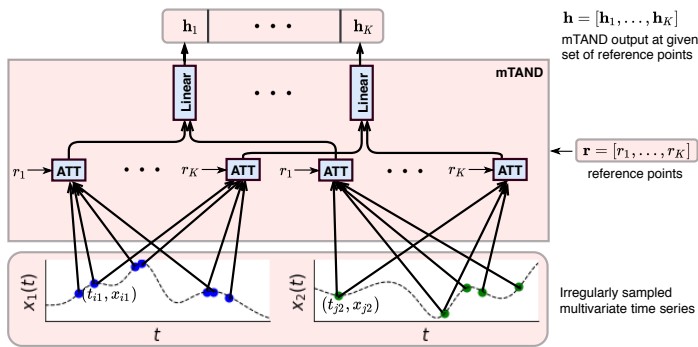

Figure 1: Architecture of the mTAND module. It takes irregularly sampled time points and corresponding values as keys and values and produces a fixed dimensional representation at the query time points. The attention blocks (**ATT**) perform a scaled dot product attention over the observed values using the time embedding of the query and key time points. Equation 3 and 4 defines this operation. Note that the output at all query points can be computed in parallel.

dimensional embedding at time $t$. This process leverages a continuous-time attention mechanism applied to the $H$ time embeddings. The complete computation is described below.

$$\text{mTAN}(t, \mathbf{s})[j] = \sum_{h=1}^{H} \sum_{d=1}^{D} \hat{x}_{hd}(t, \mathbf{s}) \cdot U_{hdj} \tag{2}$$

$$\hat{x}_{hd}(t, \mathbf{s}) = \sum_{i=1}^{L_d} \kappa_h(t, t_{id}) \, x_{id} \tag{3}$$

$$\kappa_h(t, t_{id}) = \frac{\exp\left(\phi_h(t)\mathbf{w}\mathbf{v}^T \phi_h(t_{id})^T / \sqrt{d_k}\right)}{\sum_{i'=1}^{L_d} \exp\left(\phi_h(t)\mathbf{w}\mathbf{v}^T \phi_h(t_{i'd})^T / \sqrt{d_k}\right)} \tag{4}$$

As shown in Equation 2, dimension $j$ of the mTAN embedding $\text{mTAN}(t, \mathbf{s})[j]$ is given by a linear combination of intermediate univariate continuous-time functions $\hat{x}_{hd}(t, \mathbf{s})$. There is one such function defined for each input data dimension $d$ and each time embedding $h$. The parameters $U_{hdj}$ are learnable linear combination weights.

As shown in Equation 3, the structure of the intermediate continuous-time function $\hat{x}_{hd}(t, \mathbf{s})$ is essentially a kernel smoother applied to the $d^{th}$ dimension of the time series. However, the interpolation weights $\kappa_h(t, t_{id})$ are defined based on a time attention mechanism that leverages time embeddings, as shown in Equation 4. As we can see, the same time embedding function $\phi_h(t)$ is applied for all data dimensions. The form of the attention mechanism is a softmax function over the observed time points $t_{id}$ for dimension $d$. The activation within the softmax is a scaled inner product between the time embedding $\phi_h(t)$ of the query time point $t$ and the time embedding $\phi_h(t_{id})$ of the observed time point, the key. The parameters $\mathbf{w}$ and $\mathbf{v}$ are each $d_r \times d_k$ matrices where $d_k \leq d_r$. We use a scaling factor $\frac{1}{\sqrt{d_k}}$ to normalize the dot product to counteract the growth in the dot product magnitude with increase in the dimension $d_k$.

Learning the time embeddings provides our model with flexibility to learn complex temporal kernel functions $\kappa_h(t, t')$. The use of multiple simultaneous time embeddings $\phi_h(t)$ and a final linear combination across time embedding dimensions and data dimensions means that the final output representation function $\text{mTAN}(t, \mathbf{s})$ is extremely flexible. Different input dimensions can leverage different time embeddings via learned sparsity patterns in the parameter tensor $U$. Information from different data dimensions can also be mixed together to create compact reduced dimensional representations. We note that all of the required computations can be parallelized using masking variables to deal with unobserved dimensions, allowing for efficient implementation on a GPU.

**Discretization:** Since the mTAN module defines a continuous function of $t$ given $\mathbf{s}$, it can not be directly incorporated into neural network architectures that expect inputs in the form of fixed-dimensional vectors or discrete sequences. However, the mTAN module can easily be adapted to

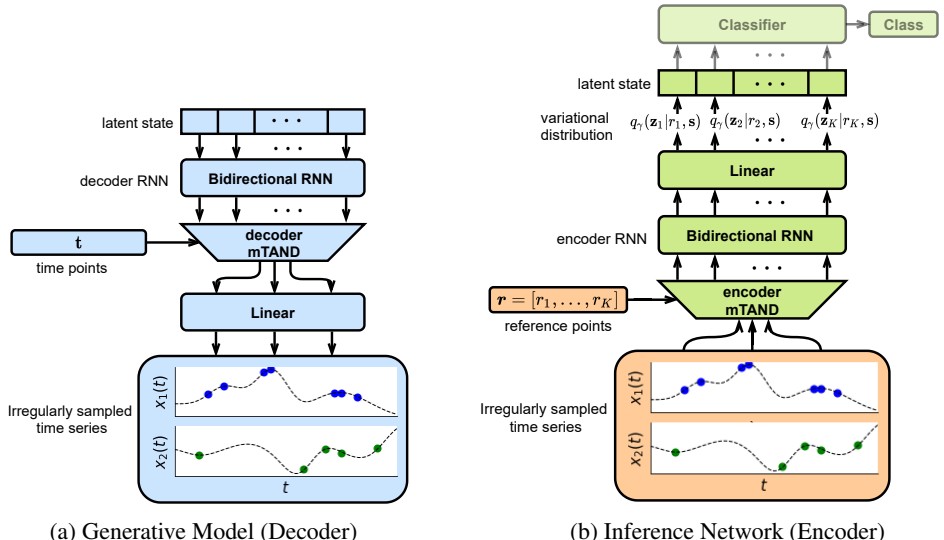

(a) Generative Model (Decoder)      (b) Inference Network (Encoder)

Figure 2: Architecture of the proposed encoder-decoder framework **mTAND-Full**. The classifier is required only for performing classification tasks. The mTAND module is shown in Figure 1.

produce such an output representation by materializing its output at a set of reference time points $\mathbf{r} = [r_1, ..., r_K]$. In some cases, we may have a fixed set of such points. In other cases, the set of reference time points may need to depend on $\mathbf{s}$ itself. In particular, we define the auxiliary function $\rho(\mathbf{s})$ to return the set of time points at which there is an observation on any dimension of $\mathbf{s}$.

Given a collection of reference time points $\mathbf{r}$, we define the discretized mTAN module mTAND$(\mathbf{r}, \mathbf{s})$ as mTAND$(\mathbf{r}, \mathbf{s})[i] = $ mTAN$(r_i, \mathbf{s})$. This module takes as input the set of reference time points $\mathbf{r}$ and the time series $\mathbf{s}$ and outputs a sequence of mTAN embeddings of length $|\mathbf{r}|$, each of dimension $J$. The architecture of the mTAN module is shown in Figure 1. The mTAN module can be used to interface sparse and irregularly sampled multivariate time series data with any deep neural network layer type including fully-connected, recurrent, and convolutional layers. In the next section, we describe the construction of a temporal encoder-decoder architecture leveraging the mTAN module, which can be applied to both classification and interpolation tasks.

## 4 ENCODER-DECODER FRAMEWORK

As described in the last section, we leverage the discretized mTAN module in an encoder-decoder framework as the primary model in this paper, which we refer to as an mTAN network. We develop the encoder-decoder framework within the variational autoencoder (VAE) framework in this section. The architecture for the model framework is shown in Figure 2.

**Model Architecture:** As we are modeling time series data, we begin by defining a sequence of latent states $\mathbf{z}_i$. Each of these latent states are IID-distributed according to a standard multivariate normal distribution $p(\mathbf{z}_i)$. We define the set of latent states $\mathbf{z} = [\mathbf{z}_1, ..., \mathbf{z}_K]$ at $K$ reference time points.

We define a three-stage decoder. First, the latent states are processed through an RNN decoder module to induce temporal dependencies resulting in a first set of deterministic latent variables $\mathbf{h}_{RNN}^{dec} = [\mathbf{h}_{1,RNN}^{dec}, ..., \mathbf{h}_{K,RNN}^{dec}]$. Second, the output of the RNN decoder stage and the $K$ time points $\mathbf{h}_{RNN}^{dec}$ are provided to the mTAND module along with a set of $T$ query time points $\mathbf{t}$. The mTAND module outputs a sequence of embeddings $\mathbf{h}_{TAN}^{dec} = [\mathbf{h}_{1,TAN}^{dec}, ..., \mathbf{h}_{T,TAN}^{dec}]$ of length $|\mathbf{t}|$. Third, the mTAN embeddings are independently decoded using a fully connected decoder $f^{dec}()$ and the result is used to parameterize an output distribution. In this work, we use a diagonal covariance Gaussian distribution with mean given by the final decoded representation and a fixed variance $\sigma^2$. The final generated time series is given by $\hat{\mathbf{s}} = (\mathbf{t}, \mathbf{x})$ with all data dimensions observed. The full generative process is shown below. We let $p_\theta(\mathbf{x}|\mathbf{z}, \mathbf{t})$ define the probability distribution over

the values of the time series $\mathbf{x}$ given the time points $\mathbf{t}$ and the latent variables $\mathbf{z}$. $\theta$ represents the parameters of all components of the decoder.

$$\mathbf{z}_k \sim p(\mathbf{z}_k) \tag{5}$$

$$\mathbf{h}_{RNN}^{dec} = \text{RNN}^{dec}(\mathbf{z}) \tag{6}$$

$$\mathbf{h}_{TAN}^{dec} = \text{mTAND}^{dec}(\mathbf{t}, \mathbf{h}_{RNN}^{dec}) \tag{7}$$

$$x_{id} \sim \mathcal{N}(x_{id}; f^{dec}(\mathbf{h}_{i,TAN}^{dec})[d], \sigma^2 \boldsymbol{I}) \tag{8}$$

For an encoder, we simply invert the structure of the generative process. We begin by mapping the input time series $\mathbf{s}$ through the mTAND module along with a collection of $K$ reference time points $\mathbf{r}$. We apply an RNN encoder to the mTAND model that outputs $\mathbf{h}_{TAN}^{enc}$ to encode longer-range temporal structure. Finally, we construct a distribution over latent variables at each reference time point using a diagonal Gaussian distribution with mean and variance output by fully connected layers applied to the RNN outputs $\mathbf{h}_{RNN}^{enc}$. The complete encoder architecture is described below. We define $q_\gamma(\mathbf{z}|\mathbf{r}, \mathbf{s})$ to be the distribution over the latent variables induced by the input time series $\mathbf{s}$ and the reference time points $\mathbf{r}$. $\gamma$ represents all of the parameters in all of the encoder components.

$$\mathbf{h}_{TAN}^{enc} = \text{mTAND}^{enc}(\mathbf{r}, \mathbf{s}) \tag{9}$$

$$\mathbf{h}_{RNN}^{enc} = \text{RNN}^{enc}(\mathbf{h}_{TAN}^{enc}) \tag{10}$$

$$\mathbf{z}_k \sim q_\gamma(\mathbf{z}_k|\boldsymbol{\mu}_k, \boldsymbol{\sigma}_k^2), \quad \boldsymbol{\mu}_k = f_\mu^{enc}(\mathbf{h}_{k,RNN}^{enc}), \quad \boldsymbol{\sigma}_k^2 = \exp(f_\sigma^{enc}(\mathbf{h}_{k,RNN}^{enc})) \tag{11}$$

**Unsupervised Learning:** To learn the parameters of our encoder-decoder model given a data set of sparse and irregularly sampled time series, we follow a slightly modified VAE training approach and maximize a normalized variational lower bound on the log marginal likelihood based on the evidence lower bound or ELBO. The learning objective is defined below where $p_\theta(x_{jdn}|\mathbf{z}, \mathbf{t}_n)$ and $q_\gamma(\mathbf{z}|\mathbf{r}, \mathbf{s}_n)$ are defined in the previous section.

$$\mathcal{L}_{\text{NVAE}}(\theta, \gamma) = \sum_{n=1}^{N} \frac{1}{\sum_d L_{dn}} \Big( \mathbb{E}_{q_\gamma(\mathbf{z}|\mathbf{r}, \mathbf{s}_n)}[\log p_\theta(\mathbf{x}_n|\mathbf{z}, \mathbf{t}_n)] - D_{\text{KL}}(q_\gamma(\mathbf{z}|\mathbf{r}, \mathbf{s}_n)||p(\mathbf{z})) \Big) \tag{12}$$

$$D_{\text{KL}}(q_\gamma(\mathbf{z}|\mathbf{r}, \mathbf{s}_n)||p(\mathbf{z})) = \sum_{i=1}^{K} D_{\text{KL}}(q_\gamma(\mathbf{z}_i|\mathbf{r}, \mathbf{s}_n)||p(\mathbf{z}_i)) \tag{13}$$

$$\log p_\theta(\mathbf{x}_n|\mathbf{z}, \mathbf{t}_n) = \sum_{d=1}^{D} \sum_{j=1}^{L_{dn}} \log p_\theta(x_{jdn}|\mathbf{z}, t_{jdn}) \tag{14}$$

Since irregularly sampled time series can have different numbers of observations across different dimensions as well as across different data cases, it can be helpful to normalize the terms in the standard ELBO objective to avoid the model focusing more on sequences that are longer at the expense of sequences that are shorter. The objective above normalizes the contribution of each data case by the total number of observations it contains. The fact that all data dimensions are not observed at all time points is accounted for in Equation 14. In practice, we use $k$ samples from the variational distribution $q_\gamma(\mathbf{z}|\mathbf{r}, \mathbf{s}_n)$ to compute the learning objective.

**Supervised Learning:** We can also augment the encoder-decoder model with a supervised learning component that leverages the latent states as a feature extractor. We define this component to be of the form $p_\delta(y_n|\mathbf{z})$ where $\delta$ are the model parameters. This leads to an augmented learning objective as shown in Equation 15 where the $\lambda$ term trades off the supervised and unsupervised terms.

$$\mathcal{L}_{\text{supervised}}(\theta, \gamma, \delta) = \mathcal{L}_{\text{NVAE}}(\theta, \gamma) + \lambda \mathbb{E}_{q_\gamma(\mathbf{z}|\mathbf{r}, \mathbf{s}_n)} \log p_\delta(y_n|\mathbf{z}) \tag{15}$$

In this work, we focus on classification as an illustrative supervised learning problem. For the classification model $p_\delta(y_n|\mathbf{z})$, we use a GRU followed by a 2-layer fully connected network. We use a small number of samples to approximate the required intractable expectations during both learning and prediction. Predictions are computed by marginalizing over the latent variable as shown below.

$$y^* = \underset{y \in \mathcal{Y}}{\arg\max} \; \mathbb{E}_{q_\gamma(\mathbf{z}|\mathbf{r}, \mathbf{s})}[\log p_\delta(y|\mathbf{z})] \tag{16}$$

## 5 EXPERIMENTS

In this section, we present interpolation and classification experiments using a range of models and three real-world data sets (Physionet Challenge 2012, MIMIC-III, and a Human Activity dataset). Additional illustrative results on synthetic data can be found in Appendix A.2.

**Datasets:** The PhysioNet Challenge 2012 dataset (Silva et al., 2012) consists of multivariate time series data with 37 variables extracted from intensive care unit (ICU) records. Each record contains sparse and irregularly spaced measurements from the first 48 hours after admission to ICU. We follow the procedures of Rubanova et al. (2019) and round the observation times to the nearest minute. This leads to 2880 possible measurement times per time series. The data set includes 4000 labeled instances and 4000 unlabeled instances. We use all 8000 instances for interpolation experiments and the 4000 labeled instances for classification experiments. We focus on predicting in-hospital mortality. 13.8% of examples are in the positive class.

The MIMIC-III data set (Johnson et al., 2016) is a multivariate time series dataset consisting of sparse and irregularly sampled physiological signals collected at Beth Israel Deaconess Medical Center from 2001 to 2012. Following the procedures of Shukla & Marlin (2019), we extract 53, 211 records each containing 12 physiological variables. We focus on predicting in-hospital mortality using the first 48 hours of data. 8.1% of the instances have positive labels.

The human activity dataset consists of 3D positions of the waist, chest and ankles collected from five individuals performing various activities including walking, sitting, lying, standing, etc. We follow the data preprocessing steps of Rubanova et al. (2019) and construct a dataset of 6, 554 sequences with 12 channels and 50 time points. We focus on classifying each time point in the sequence into one of eleven types of activities.

**Experimental Protocols:** We conduct interpolation experiments using the 8000 data cases in the PhysioNet data set. We randomly divide the data set into a training set containing 80% of the instances, and a test set containing the remaining 20% of instances. We use 20% of the training data for validation. In the interpolation task, we condition on a subset of available points and predict values for rest of the time points. We perform interpolation experiments with a varying percentage of observed points ranging from 50% to 90% of the available points. At test time, the values of observed points are conditioned on and each model is used to infer the values at rest of the available time points in the test instance. We repeat each experiment five times using different random seeds to initialize the model parameters. We assess performance using mean squared error (MSE).

We use the labeled data in all three data sets to conduct classification experiments. The PhysioNet and MIMIC III problems are whole time series classification problems. Note that for the human activity dataset, we classify each time point in the time series. We treat this as a smoothing problem and condition on all available observations when producing the classification at each time-point (similar to labeling in a CRF). We use bidirectional RNNs as the RNN-based baselines for the human activity dataset. We randomly divide each data set into a training set containing 80% of the time series, and a test set containing the remaining 20% of instances. We use 20% of the training set for validation. We repeat each experiment five times using different random seeds to initialize the model parameters. Due to class imbalance in the Physionet and MIMIC-III data sets, we assess classification performance using area under the ROC curve (the AUC score). For the Human Activity dataset, we evaluate models using accuracy.

For both interpolation and prediction, we select hyper-parameters on the held-out validation set using grid search, and then apply the best trained model to the test set. The hyper-parameter ranges searched for each model/dataset/task are fully described in Appendix A.4.

**Models:** The model we focus on is the encoder-decoder architecture based on the discretized multi-time attention module (**mTAND-Full**). In the classification experiments, the hidden state at the last observed point is passed to a two-layer binary classification module for all models. For each data set, the structure of this classifier is the same for all models. For the proposed model, the sequence of latent states is first passed through a GRU and then the final hidden state is passed through the same classification module. For the classification task only, we consider an ablation of the full model that uses the proposed mTAND encoder, which consists of our mTAND module followed by a GRU to extract a final hidden state, which is then passed to the classification module (**mTAND-Enc**). We compare to several deep learning models that expand on recurrent networks to accommodate irregular

Table 1: Interpolation performance versus percent observed time points on PhysioNet

| Model | Mean Squared Error ($\times 10^{-3}$) | | | | |
|---|---|---|---|---|---|
| RNN-VAE | $13.418 \pm 0.008$ | $12.594 \pm 0.004$ | $11.887 \pm 0.005$ | $11.133 \pm 0.007$ | $11.470 \pm 0.006$ |
| L-ODE-RNN | $8.132 \pm 0.020$ | $8.140 \pm 0.018$ | $8.171 \pm 0.030$ | $8.143 \pm 0.025$ | $8.402 \pm 0.022$ |
| L-ODE-ODE | $6.721 \pm 0.109$ | $6.816 \pm 0.045$ | $6.798 \pm 0.143$ | $6.850 \pm 0.066$ | $7.142 \pm 0.066$ |
| mTAND-Full | $\mathbf{4.139 \pm 0.029}$ | $\mathbf{4.018 \pm 0.048}$ | $\mathbf{4.157 \pm 0.053}$ | $\mathbf{4.410 \pm 0.149}$ | $\mathbf{4.798 \pm 0.036}$ |
| Observed % | 50% | 60% | 70% | 80% | 90% |

sampling. We also compare to several encoder-decoder approaches. The full list of model variants is briefly described below. We use a Gated Recurrent Unit (GRU) (Chung et al., 2014) module as the recurrent network throughout. Architecture details can be found in Appendix A.3.

- **RNN-Impute:** Missing observations replaced with weighted average of last observed measurement within that time series and global mean of the variable across training examples (Che et al., 2018).

- **RNN-$\Delta_t$:** Input is concatenated with masking variable and time interval $\Delta_t$ indicating how long the particular variable is missing.

- **RNN-Decay:** RNN with exponential decay on hidden states (Mozer et al., 2017; Che et al., 2018).

- **GRU-D:** combining hidden state decay with input decay (Che et al., 2018).

- **Phased-LSTM:** Captures time irregularity by a time gate that regulates access to the hidden and cell state of the LSTM (Neil et al., 2016) with forward filling to handle partially observed vectors.

- **IP-Nets:** Interpolation prediction networks, which use several semi-parametric RBF interpolation layers, followed by a GRU (Shukla & Marlin, 2019).

- **SeFT:** Uses a set function based approach where all the observations are modeled individually before pooling them together using an attention based approach (Horn et al., 2020).

- **RNN-VAE:** A VAE-based model where the encoder and decoder are standard RNN models.

- **ODE-RNN:** Uses neural ODEs to model hidden state dynamics and an RNN to update the hidden state in presence of a new observation (Rubanova et al., 2019).

- **L-ODE-RNN:** Latent ODE where the encoder is an RNN and decoder is a neural ODE (Chen et al., 2018).

- **L-ODE-ODE:** Latent ODE where the encoder is an ODE-RNN and decoder is a neural ODE (Rubanova et al., 2019).

**Physionet Experiments:** Table 1 compares the performance of all methods on the interpolation task where we observe $50\% - 90\%$ of the values in the test instances. As we can see, the proposed method (mTAND-Full) consistently and substantially outperforms all of the previous approaches across all of the settings of observed time points. We note that in this experiment, different columns correspond to different setting (for example, in the case of 70%, we condition on 70% of data and predict the rest of the data; i.e., 30%) and, hence the results across columns are not comparable.

Table 2 compares predictive performance on the PhysioNet mortality prediction task. The full Multi-Time Attention network model (mTAND-Full) and the classifier based only on the Multi-Time Attention network encoder (mTAND-Enc) achieve significantly improved performance relative to the current state-of-the-art methods (ODE-RNN and L-ODE-ODE) and other baseline methods.

We also report the time per epoch in minutes for all the methods. We note that the ODE-based models require substantially more run time than other methods due to the required use of an ODE solver (Chen et al., 2018; Rubanova et al., 2019). These methods also require taking the union of all observation time points in a batch, which further slows down the training process. As we can see, the proposed full Multi-Time Attention network (mTAND-Full) is over 85 times faster than ODE-RNN and over 100 times faster than L-ODE-ODE, the best-performing ODE-based models.

**MIMIC-III Experiments:** Table 2 compares the predictive performance of the models on the mortality prediction task on MIMIC-III. The Multi-Time Attention network-based encoder-decoder framework (mTAND-Full) achieves better performance than the recent IP-Net and SeFT model as well as all of the RNN baseline models. While ODE-RNN and L-ODE-ODE both have slightly better

Table 2: Classification Performance on PhysioNet, MIMIC-III and Human Activity dataset

| Model | AUC Score | | Accuracy | time |
| --- | --- | --- | --- | --- |
| | **PhysioNet** | **MIMIC-III** | **Human Activity** | **per epoch** |
| RNN-Impute | $0.764 \pm 0.016$ | $0.8249 \pm 0.0010$ | $0.859 \pm 0.004$ | 0.5 |
| RNN-$\Delta_t$ | $0.787 \pm 0.014$ | $0.8364 \pm 0.0011$ | $0.857 \pm 0.002$ | 0.5 |
| RNN-Decay | $0.807 \pm 0.003$ | $0.8392 \pm 0.0012$ | $0.860 \pm 0.005$ | 0.7 |
| RNN GRU-D | $0.818 \pm 0.008$ | $0.8270 \pm 0.0010$ | $0.862 \pm 0.005$ | 0.7 |
| Phased-LSTM | $0.836 \pm 0.003$ | $0.8429 \pm 0.0035$ | $0.855 \pm 0.005$ | 0.3 |
| IP-Nets | $0.819 \pm 0.006$ | $0.8390 \pm 0.0011$ | $0.869 \pm 0.007$ | 1.3 |
| SeFT | $0.795 \pm 0.015$ | $0.8485 \pm 0.0022$ | $0.815 \pm 0.002$ | 0.5 |
| RNN-VAE | $0.515 \pm 0.040$ | $0.5175 \pm 0.0312$ | $0.343 \pm 0.040$ | 2.0 |
| ODE-RNN | $0.833 \pm 0.009$ | $\mathbf{0.8561 \pm 0.0051}$ | $0.885 \pm 0.008$ | 16.5 |
| L-ODE-RNN | $0.781 \pm 0.018$ | $0.7734 \pm 0.0030$ | $0.838 \pm 0.004$ | 6.7 |
| L-ODE-ODE | $0.829 \pm 0.004$ | $\mathbf{0.8559 \pm 0.0041}$ | $0.870 \pm 0.028$ | 22.0 |
| mTAND-Enc | $0.854 \pm 0.001$ | $0.8419 \pm 0.0017$ | $\mathbf{0.907 \pm 0.002}$ | **0.1** |
| mTAND-Full | $\mathbf{0.858 \pm 0.004}$ | $0.8544 \pm 0.0024$ | $\mathbf{0.910 \pm 0.002}$ | 0.2 |

mean AUC than mTAND-Full, the differences are not statistically significant. Further, as shown on the PhysioNet classification problem, mTAND-Full is more than an order of magnitude faster than the ODE-based methods.

**Human Activity Experiments:** Table 2 shows that the mTAND-based classifiers achieve significantly better performance than the baseline models on this prediction task, followed by ODE-based models and IP-Nets.

**Additional Experiments:** In Appendix A.2, we demonstrate the effectiveness of learning temporally distributed latent representations with mTANs on a synthetic dataset. We show that mTANs are able to capture local structure in the time series better than latent ODE-based methods that encode to a single time point. This property of mTANs helps to improve the interpolation performance in terms of mean squared error.

We also perform ablation experiments to show the performance gain achieved by learning similarity kernels and time embeddings in Appendix A.1. In particular, we show that learning the time embedding improves classification performance compared to using fixed positional encodings. We also demonstrate the effectiveness of learning the similarity kernel by comparing to an approach that uses fixed RBF kernels. Appendix A.1 shows that learning the similarity kernel using the mTAND module performs as well as or better than using a fixed RBF kernel.

## 6 DISCUSSION AND CONCLUSIONS

In this paper, we have presented the Multi-Time Attention (mTAN) module for learning from sparse and irregularly sampled data along with a VAE-based encoder-decoder model leveraging this module. Our results show that the resulting model performs as well or better than a range of baseline and state-of-the-art models on both the interpolation and classification tasks, while offering training times that are one to two orders of magnitude faster than previous state of the art methods. While in this work we have focused on a VAE-based encoder-decoder architecture, the proposed mTAN module can be used to provide an interface between sparse and irregularly sampled time series and many different types of deep neural network architectures including GAN-based models. Composing the mTAN module with convolutional networks instead of recurrent architectures may also provide further computational enhancements due to improved parallelism.

## ACKNOWLEDGEMENTS

Research reported in this paper was partially supported by the National Institutes of Health under award numbers 5U01CA229445 and 1P41EB028242.

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

# A   APPENDIX

## A.1   ABLATION STUDY

In this section, we perform ablation experiments to show the performance gain achieved by learning similarity kernel and time embedding. Table 3 shows the ablation results by substituting fixed positional encoding (Vaswani et al., 2017) in place of learnable time embedding defined in Equation 1 in mTAND-Full model on PhysioNet and MIMIC-III dataset for classification task. We report the average AUC score over 5 runs. As we can see from Table 3, learning the time embedding improves AUC score by 1% as compared to using fixed positional encodings.

Table 3: Ablation with time embedding

| Dataset | Time Embedding | AUC Score |
|---------|----------------|-----------|
| PhysioNet | Positional Encoding | $0.845 \pm 0.004$ |
|  | Learned Time Embedding | $\mathbf{0.858 \pm 0.004}$ |
| MIMIC-III | Positional Encoding | $0.843 \pm 0.001$ |
|  | Learned Time Embedding | $\mathbf{0.854 \pm 0.002}$ |

Since mTANs are fundamentally continuous-time interpolation-based models, we perform an ablation study by comparing mTANs with the IP-nets (Shukla & Marlin, 2019). IP-Nets use several semi-parametric RBF interpolation layers, followed by a GRU to model irregularly sampled time series. In this framework, we replace the RBK kernel with a learnable similarity kernel using mTAND module, the corresponding model is mTAND-Enc. Table 4 compares the performance of the two methods on classification task on PhysioNet, MIMIC-III and Human Activity dataset. We report the average AUC score over 5 runs. Table 4 shows that learning the similarity kernel using mTAND module performs as well or better than using a fixed RBF kernel.

Table 4: Comparing interpolation kernels

| Dataset | Model | AUC Score |
|---------|-------|-----------|
| PhysioNet | IP-Nets | $0.819 \pm 0.006$ |
|  | mTAND-Enc | $\mathbf{0.854 \pm 0.001}$ |
| MIMIC-III | IP-Nets | $\mathbf{0.839 \pm 0.001}$ |
|  | mTAND-Enc | $\mathbf{0.842 \pm 0.001}$ |
| Human Activity | IP-Nets | $0.869 \pm 0.007$ |
|  | mTAND-Enc | $\mathbf{0.907 \pm 0.002}$ |

## A.2   SYNTHETIC INTERPOLATION EXPERIMENTS

To demonstrate the capabilities of our model on the interpolation task, we generate a synthetic dataset consisting of 1000 trajectories each of 100 time points sampled over $t \in [0, 1]$. We fix 10 reference points and use RBF kernel with a fixed bandwidth of 100 for constructing local interpolations at 100 time points over $[0, 1]$. The values at the reference points are drawn from a standard normal distribution.

We randomly sample 20 observations from each trajectory to simulate a sparse and irregularly sampled multivariate time series. We use 80% of the data for training and 20% for testing. At test time, encoder conditions on 20 irregularly sampled time points and the decoder generates interpolations on all 100 time points. Figure 3 illustrates the interpolation results on the test set for the Multi-Time Attention Network and Latent ODE model with ODE encoder (Rubanova et al., 2019). For both the models, we draw 100 samples from the approximate posterior distribution. As we can see from Figure 3, the ODE interpolations are much smoother and haven't been able to capture the local structure as well as mTANS.

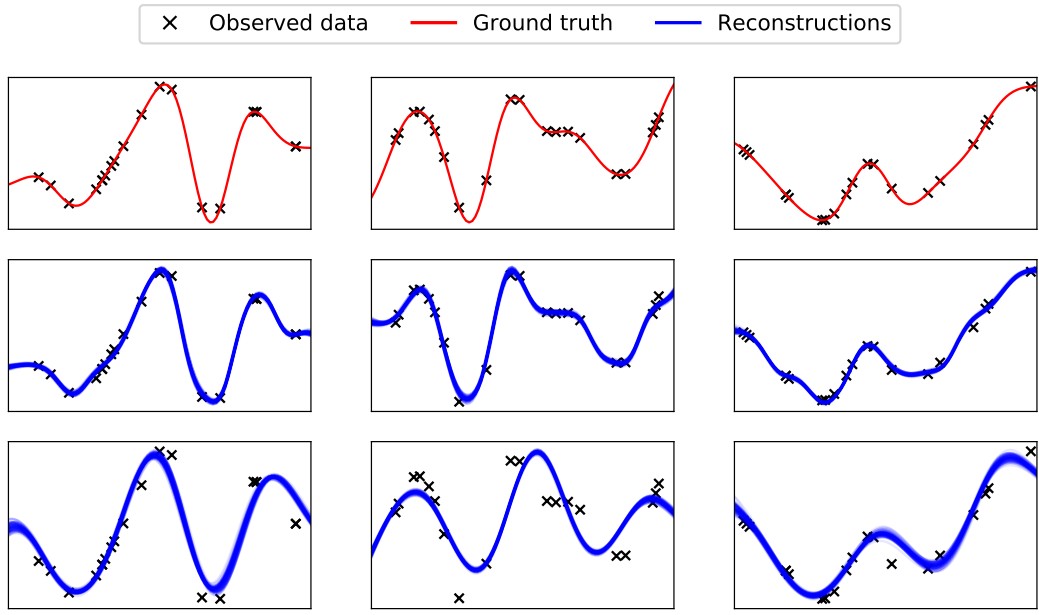

Figure 3: Interpolations on the synthetic interpolation dataset. The columns represent 3 different examples. First row: Ground truth trajectories with observed points, second row: reconstructions on the complete range $t \in [0, 1]$ using the proposed model mTAN, third row: reconstructions on the complete range $t \in [0, 1]$ using the Latent ODE model with ODE encoder.

Table 5: Synthetic Data: Mean Squared Error

| Latent Dimension | Model | Reconstruction | Interpolation |
|---|---|---|---|
| 10 | L-ODE-ODE | 0.0209 | 0.0571 |
| | **mTAND-Full** | **0.0088** | **0.0409** |
| 20 | L-ODE-ODE | 0.0191 | 0.0541 |
| | **mTAND-Full** | **0.0028** | **0.0335** |

Table 5 compares the proposed model with best performing baseline Latent-ODE with ODE encoder (L-ODE-ODE) on reconstruction and interpolation task. For both the tasks, we condition on the 20 irregularly sampled time points and reconstruct the input points (reconstruction) and the whole set of 100 time points (interpolation). We report the mean squared error on test set.

## A.3 ARCHITECTURE DETAILS

**Multi-Time Attention Network (mTAND-Full)**: In our proposed encoder-decoder framework (Figure 2), we use bi-directional GRU as the recurrent model in both encoder and decoder. In encoder, we use a 2 layer fully connected network with 50 hidden units and ReLU activations to map the RNN hidden state at each reference point to mean and variance. Similarly in decoder, mTAN embeddings are independently decoded using a 2 layer fully connected network with 50 hidden units and ReLU activations, and the result is used to parameterize the output distribution. For classification tasks, we use a separate GRU layer on top of the latent states followed by a 2-layer fully connected layer with 300 units and ReLU activations to output the class probabilities.

**Multi-Time Attention Encoder (mTAND-Enc)**: As we show in the experiments, the proposed mTAN module can standalone be used for classification tasks. The mTAND-Enc consists of Multi-Time attention module followed by GRU to extract the final hidden state which is then passed to a 2-layer fully connected layer to output the class probabilities.

**Loss Function:** For computing the evidence lower bound (ELBO) during training, we use negative log-likelihood with fixed variance as the reconstruction loss. For all the datasets, we use a fixed variance of $0.01$. For computing ELBO, we use 5 samples for interpolation task and 1 sample for classification tasks. We use cross entropy loss for classification. For the classification tasks, we tune the $\lambda$ parameter in the supervised learning loss function (Equation 15). We achieved best performance using $\lambda$ as 100 and 5 for Physionet, MIMIC-III respectively. For human activity dataset, we achieved best results without using the regulaizer or ELBO component. We found that KL annealing with coeff $0.99$ improved the performance of interpolation and classification tasks on Physionet.

## A.4 HYPERPARAMETERS

**Baselines:** For Physionet and Human Activity dataset, we use the reported hyperparameters for RNN baselines as well as ODE models from Rubanova et al. (2019). For MIMIC-III dataset, we independently tune the hyperparameters of the baseline models on the validation set. We search for GRU hidden units, latent dimension, number of hidden units in fully connected network for ODE function in recognition and generative model over the range $\{20, 32, 64, 128, 256\}$. For ODEs, we also searched the number of layers in fully connected network in the range $\{1, 2, 3\}$.

**mTAN:** We learn time embeddings of size $128$. The number of embeddings $H \in \{1, 2, 4\}$. The linear projection matrices used for projecting time embedding $W$ are each $d_k * d_k/h$ where $d_k$ is the embedding size. We search the latent dimension and GRU encoder hidden size over the range $\{32, 64, 128\}$. We keep GRU decoder hidden size at 50. For the classification tasks, we use 128 reference points. For interpolation task, we search number of reference points over the range $\{8, 16, 32, 64, 128\}$. We use Adam Optimizer for training the models. For classification, experiments are run for 300 iteration with learning rate $0.0001$, while for interpolation task experiments are run for 500 iterations with learning rate $0.001$. Best hyperparameters are reported in the code.

## A.5 VISUALIZING ATTENTION WEIGHTS

Regularly sampled time points

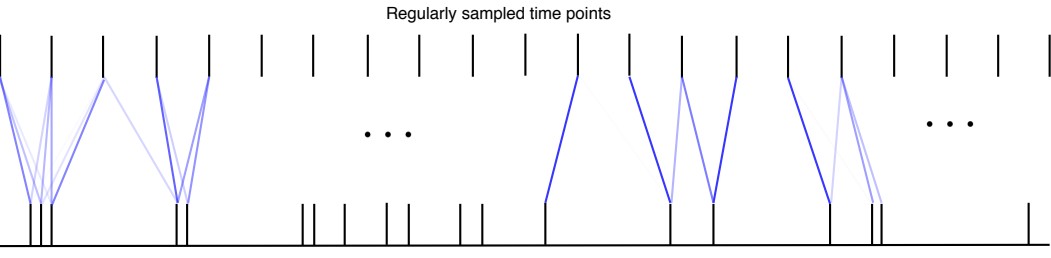

Irregularly sampled time points

Figure 4: Visualization of attention weights. mTAN learns an interpolation over the query time points by attending to the observed values at key time points. The brighter edges correspond to higher attention weights.

In this section, we visualize the attention weights learned by our proposed model. We experiment using synthetic dataset (described in A.2) which consists of univariate time series. Figure 4 shows the attention weights learned by the encoder mTAND module. The input shown in the figure is the irregularly sampled time points and the edges show how the output at reference points attends to the values on the input time points. The final output can be computed by substituting the attention weights in Equation 3.

## A.6 TRAINING DETAILS

### A.6.1 DATA GENERATION AND PREPROCESSING

All the datasets used in the experiments are publicly available and can be downloaded using the following links:
PhysioNet: `https://physionet.org/content/challenge-2012/`
MIMIC-III: `https://mimic.physionet.org/`

Human Activity: `https://archive.ics.uci.edu/ml/datasets/Localization+Data+for+Person+Activity`.

We rescale each feature to be between $0$ and $1$ for Physionet and MIMIC-III dataset. We also rescale the time to be in $[0, 1]$ for all datasets. In case of MIMIC-III dataset, for the time series missing entirely, we follow the preprocessing steps of Shukla & Marlin (2019) and assign the starting point (time t=0) value of the time series to the global mean for that variable.

### A.6.2 SOURCE CODE

The code for reproducing the results in this paper is available at `https://github.com/reml-lab/mTAN`.

### A.6.3 COMPUTING INFRASTRUCTURE

All experiments were run on a Nvidia Titan X GPU.

