# OpenReview forum: "Multi-Time Attention Networks for Irregularly Sampled Time Series"
_ICLR.cc/2021/Conference — ICLR 2021 Poster_

### Official Review · AnonReviewer2 · 2020-10-23
**A novel approach and interesting problem, but paper can be further polished.**

**Rating:** 7
**Confidence:** 4

**Review:**

This paper proposed a new model (mTANs) for sparse and irregularly sampled multivariate time series. It incorporates the time attention mechanism to learn embedding for continuous time-series based on a kernel smoothing method. Results on real-world dataset such as EHR data has outperformed other baselines.

I have the following comments that I think is worthwhile to consider to improve the paper:

1. I didn't see much on the main method sections that specifically address/discuss sparsity and irregular sample problems, it seems like the proposed method also works for general time-series data.

2. In the discretization part, it would be better to make it clear how to find the set of reference time points r in practice. It may not be efficient enough if one needs to track each time stamp for each variable to obtain this set $r$.

3. How does an extra module of mTAND in encoding and decoding procedure help intuitively? Given that RNN has already captured temporal information.

4. The noise distribution for sparse / irregularly sampled data can often be heavily skewed, so the Gaussian noise assumption in this paper may not hold.

Additional questions/suggestions:
1. It is better to specify how to compute the gradient for ELBO, e.g., are there any approximations used, are methods like REINFORCE or Gumbel softmax implemented for non-continuous cases, as well as how does the parameter initialize for unsupervised learning problems.

2. Does the unified supervised and unsupervised objective in Eq (15) mean the proposed method combines imputation/interpolation with learning/inference for time series data?

3. From the time series perspective, the positional encoding in Eq (1) is more like a season and trend decomposition, not sure if this plays a similar role here.

4. It would be very helpful for me to understand the paper if the authors could add a schematic figure to demonstrate the complete structure of the model.

---

> ### Author Response · Authors · 2020-11-13
> **Response to Reviewer 2**
>
> Thank you for your helpful comments. We address the issues below:
>
> *Q: I didn't see much on the main method sections that specifically address/discuss sparsity and irregular sample problems, it seems like the proposed method also works for general time series data.*
>
> A: As shown in Eq 3, the intermediate continuous-time function $\hat{x}_{hd}(t, \mathbf{s})$ computed at query point $t$ is only based on the observed time points
>
> $\mathbf{t_d} = [t_{1d}, t_{2d}, \cdots]$ and the corresponding values  $\mathbf{x_d}=[x_{1d},x_{2d}, \cdots]$. Irrespective of time series being sparse and irregularly sampled or regularly sampled, the flexibility in Eq 3 allows the intermediate output to be computed at any set of query points and with any set of input time points and values. We will clarify this in the paper.
>
> *Q:  In the discretization part, it would be better to make it clear how to find the set of reference time points  r in practice. It may not be efficient enough if one needs to track each time stamp for each variable to obtain this set r.*
>
> A: Assuming that all data cases are defined over a common interval, which is the case in all of our experiments, the set of reference time points $r$ is chosen to be $T$ evenly spaced time points within that interval. The number of such points is a hyper-parameter of the method.
>
> *Q: How does an extra module of mTAND in encoding and decoding procedure help intuitively? Given that RNN has already captured temporal information.*
>
> A: Standard RNNs do not represent time at all and require fully observed sequences of vector-valued inputs. These are reasonable assumptions for regularly spaced, completely observed multivariate time series, but not for irregularly sampled and unaligned multivariate time series where different dimensions can be observed at different time points for the same data case and there are variable time gaps between observations. These problems can be addressed with extensions of standard RNN models like GRU-D and ODE-RNN, which are exactly the models we compare to. Relative to models like GRU-D that operate forward in time only, the mTAND module when used as an encoder looks at the whole input time series via attention and produces a regularly spaced sequence of hidden representations. A standard RNN can then be applied to extract any additional structure from the sequence of regularly spaced hidden values produced by the  mTAND module. When used as a decoder, the mTAND module is similarly able to reconstruct points that are not aligned or regularly spaced.
>
> *Q: The noise distribution for sparse / irregularly sampled data can often be heavily skewed, so the Gaussian noise assumption in this paper may not hold.*
>
> A: The Gaussian noise distribution applies at the output of a deep generator. This is standard for many VAE architectures. Any other distribution over continuous random variables could also be used instead if desired, as is the case for other VAEs.
>
> *Q: It is better to specify how to compute the gradient for ELBO, e.g., are there any approximations used, are methods like REINFORCE or Gumbel softmax implemented for non-continuous cases, as well as how does the parameter initialized for unsupervised learning problems.*
>
> A: We are not dealing with discrete latent variables in this work. No additional approximations are required other than those commonly used in VAEs with Gaussian latent variables. We use $k$ samples to compute the ELBO and follow the reparameterization trick (Kingma & Welling, 2014) for computing the gradients.
>
> *Q: Does the unified supervised and unsupervised objective in Eq (15) mean the proposed method combines imputation/interpolation with learning/inference for time series data?*
>
> A: As shown in Equation 15, the supervised learning component leverages the latent states as a feature extractor and the unsupervised loss (ELBO) acts as a regularizer. Although this question is not entirely clear, we want to clarify that the supervised learning model does not leverage the final imputed output from the decoder as input.
>
> *Q: It would be very helpful for me to understand the paper if the authors could add a schematic figure to demonstrate the complete structure of the model.*
>
> A: We have added an architecture diagram of our proposed encoder-decoder framework and mTAND module in Appendix A.3 (Figure 2 and 3).

---

### Official Review · AnonReviewer4 · 2020-10-28
**Official Blind Review**

**Rating:** 7
**Confidence:** 4

**Review:**

The paper proposes a novel deep learning framework for handling irregularly sampled time series.
The paper is well-written and easy to follow.
The key idea of the paper is to learn embeddings for continuous time values and leveraging a time attention mechanism to learn temporal similarity from data instead of using fixed kernels.
The time embedding component takes a continuous time point and embeds it into multiple fixed-dimensional spaces. The multi-time attention mechanism takes a query time t and a multivariate sparse and irregularly sampled time series, and returns a fixed dimensional embedding for the query time t. This mechanism is used twice in an encoder-decoder VAE framework with varying reference (input) and query (output) time points.

The idea of the paper is novel, impactful, and well-explained.
The evaluation and benchmarking is proper with significant improvements over the baselines for classification and interpolation tasks. The approach has significant computational gains over the best performing baseline making it more useful in practice while achieving better or similar classification/interpolation performance.

One query I had is regarding the application of the proposed framework for extrapolation or forecasting tasks. Is the framework directly applicable to such tasks given the way time is handled to get the embeddings as the 0th dimension would keep growing with time (Eqn. 1)? This can have practical implications or challenges even in variable length classification tasks where longer duration time series can be present at test time.

Typos:
time series data data
with mroe than one
In sec 4, input and the generated time series are both denoted by vector s.

---

> ### Author Response · Authors · 2020-11-19
> **Thank you for the positive review!**
>
> Many thanks for your strong review, we’re glad to hear you are pleased with the paper!
>
> *Q: One query I had is regarding the application of the proposed framework for extrapolation or forecasting tasks.*
>
> A: In this work, we have focused on whole time series classification/regression and interpolation like problems.  It is unclear if the time embeddings defined here would work for forecasting task and we leave this for future work.
>
> *Q: typos...*
>
> A: Thanks for your detailed reading. We have fixed all the typos.

---

### Official Review · AnonReviewer3 · 2020-10-30
**Multi-Time Attention Networks for Irregularly Sampled Time Series**

**Rating:** 6
**Confidence:** 4

**Review:**

This paper proposes a novel approach to learn an embedding of continuous time values and use an attention mechanism to produce a fixed-length representation of a time series containing a variable number of observations. In particular, it proposes an mTAN network to leverage the mTAN module in an encoder-decoder framework for both unsupervised and supervised Learning.  The main contribution of this paper is the introduction of Multi-Time Attention Networks to learns a time representation and learns to attend to observations at different time points by computing a similarity weighting by the learning time embedding.  Empirical studies are performed to show the superiority of the proposed model mTANs over several baseline approaches on the tasks unsupervised and supervised learning.

Pros:

1. The paper proposes a novel model to learns a continuous time representation and adapt to fixed dimensional vectors or discrete sequences. For me, the problem itself is real and practical.
2. The proposed mTAN is novel for capturing the time dependencies time-series, sparse, irregularly sampled, and multivariate data.
3. This paper provides comprehensive experiments, including both unsupervised and supervised learning results, to show the effectiveness of the proposed framework.

Cons:

1.	The paper uses a lot of notations in equations and descriptions, which cause a little bit confusion to follow the authors’ idea. Please consider providing a table to list all the import symbols.
2.	It is better to depict this proposed model structure give the audience a main picture to model.
3.	As the model is attention-based, it has the ability to find the relationships among sequential events. Is it possible to provide one or two case studies to demonstrate the dependencies between time points?


Questions during rebuttal period:

Please address and clarify the cons above

Minor comments:
1.A few references only list authors, title, and year, but miss publisher or conference, such as “ Michael Mozer, Denis Kazakov, and Robert Lindsey. Discrete event, continuous time rnns. 2017“

---

> ### Author Response · Authors · 2020-11-18
> **Response to Reviewer 3**
>
> Thank you for your helpful comments. We address the issues below:
>
> *Q: The paper uses a lot of notations in equations and descriptions, which cause a little bit confusion to follow the authors’ idea. Please consider providing a table to list all the import symbols.*
>
> A: We have added a table of notations in Appendix A.7 for better readability of the equations.
>
> *Q: It is better to depict this proposed model structure give the audience a main picture to model.*
>
> A: We have added an architecture diagram of our proposed encoder-decoder framework and mTAND module in Appendix A.3 (Figure 2 and 3).
>
> *Q: As the model is attention-based, it has the ability to find the relationships among sequential events. Is it possible to provide one or two case studies to demonstrate the dependencies between time points?*
>
> A: We have added a visualization of attention weights in Appendix A.5 on a synthetic data example.
>
> *Q: A few references only list authors, title, and year, but miss publisher or conference...*
>
> A: Thanks for pointing this out. We have fixed all the references that were missing publisher or conference.

---

### Official Review · AnonReviewer1 · 2020-10-30
**Results push us to accept this article but the lack of clarity in the model description is a strong weak point**

**Rating:** 7
**Confidence:** 3

**Review:**

This article tackles the analysis of irregular samples time series. The approach is mainly based on interpolation. Thus, the authors can del with both supervised and unsupervised problems.
The architecture is made of a sinusoid attention layer, a VAE layer that lead to a fixed size set of landmark in the latent space and a RNN decoder.
For the supervised task, the authors add a classification loss.

They obtain impressive results on the interpolation task and interesting results on the classification task.

* In an interpolation problem, we would like to consider a robust baseline as a linear interpolation or an AR-like modeling. Even if I must admit that the authors already propose a lot of comparisons with models from the -recent- litterature, this would give us a meaningful MSE result to compare other approaches.

* notations should be improved (and/or completed with a schema). The model is really very difficult to understand in this version of the article.

* Results are impressive but I don't get which part of the architecture lead to such a performance

---

> ### Author Response · Authors · 2020-11-18
> **Thank you for the positive review!**
>
> Thank you for the positive review. We respond to your comments below:
>
> *Q: notations should be improved (and/or completed with a schema). The model is really very difficult to understand in this version of the article.*
>
> A: We have added an architecture diagram of our proposed encoder-decoder framework and mTAND module in Appendix A.3 (Figure 2 and 3).  We have also  added a table of notations in Appendix A.7 for better readability of the equations. We hope that this makes our proposed framework easy to understand.
>
> *Q: Results are impressive but I don't get which part of the architecture lead to such a performance.*
>
> A: We have performed ablation experiments in Appendix A.1 to show the performance gain achieved by learning similarity kernel and time embedding.  Since mTANs are fundamentally continuous-time interpolation-based models, we perform an ablation study by comparing mTAND-based similarity kernel with the IP-nets (Shukla & Marlin, 2019), which use RBF-kernel based interpolation layers. We show that learning the similarity kernel using mTAND module performs better than using a fixed RBF kernel on classification tasks. Learning the similarity function provides our model with flexibility to learn complex temporal kernel functions and hence improves over methods using fixed similarity kernel.
>
> *In an interpolation problem, we would like to consider a robust baseline as a linear interpolation or an AR-like modeling. Even if I must admit that the authors already propose a lot of comparisons with models from the -recent- literature, this would give us a meaningful MSE result to compare other approaches.*
>
> A: We have compared the performance of our proposed model to simple interpolation-based methods on interpolation task. We perform this experiment on synthetic dataset and use the same setting of the experiment described in Appendix A.2.
>
> -----------------------------------
> Method          |   MSE
>
> mTAND-Full   |   0.0335
>
> MICE               |   0.1634
>
> Linear             |   0.0909
>
> Previous         |   0.1129
>
> ------------------------------------
> Linear: linear interpolation, Previous: interpolating using previous value, MICE: multivariate imputations by chained equations

---

### Decision · Program_Chairs · 2021-01-07
**Final Decision**

**Decision:**

Accept (Poster)

**Comment:**

The work proposed a new approach to encode time series that are irregularly sampled and multivariate using time attention module and an encoder-decoder framework based on VAE. All the reviewers find the approach novel and the experiments extensive with encouraging results. Please continue to improve the presentation of the paper. I would  suggest to move the diagram showing the overall architecture to the main text to assist the explanation. Reviewers also would like to see more explanation on the experimental results and some ablation studies to show the importance of each component of the proposed architecture.